[Reviews · NeurIPS 2017]

Reviewer 1



This paper uses a reinforcement learning approach to equip a sequence-to-sequence lstm NMT model with an objective prediction model so that the greedy beam search that optimizes maximum likelihood is tempered by a prediction of the objective based on the current state of the search. The paper is well written and clearly outlines the model design and experimental procedure. The data and comparisons are all apt, the baselines are chosen wisely and are not weak, and the result is quite significant; I will probably incorporate these results in my own NMT research. I would, however, like to see a comparison between this work and the work of the cited Shen et al. paper that directly trains NMT to maximize BLEU instead of maximum likelihood. It seems that the same information is being learned in this work and, if so, that would be a more efficient way to learn. A difference between the two would tease apart the impact of the RL vs the different objective. That work does evaluate on 2 of the sets shown here; I did not carefully check to see that the entire experimental setup is the same, but if so, then you could easily use those results to strengthen your case. I was confused about one aspect of training: sentence-level BLEU is used as an objective to train the Value NN, but that metric, without any modifications, can be quite erratic when used to evaluate a single sentence, since 0 4-gram matches will render the score 0 and not take into account partial matches. Various tricks have been employed to get around this; BLEU+1 is the most common (see http://www3.nd.edu/~dchiang/papers/mira.pdf or http://www.aclweb.org/anthology/D07-1080). How was this handled? 51&52: outputted->output 81: proceeding->preceding eq 3 and 4: g and q functions not defined 96: 30k^50 candidates. it's sum_{i=1}^50 30k^i candidate *states* 104-113: long winded. You could just say, essentially, "Beam decoding makes search errors" if you need to make space.

Reviewer 2



This paper addresses one of the limitation of NMT, the so-called exposure bias, that results from the fact that each word is chosen greedily. For this, the authors build on standard technique of reinforcement learning and try to predict, for each outgoing transition of a given state, the expected reward that will be achieved if the system take this transition. The article is overall very clear and the proposed ideas quite appealing, even if many of the decisions seem quite ad hoc (e.g. why doing a linear combination between the (predicted) expected reward and the prediction of the NMT system and not decoding directly with the former) and it is not clear whether the proposed approach can be applied to other NLP tasks. More importantly, several implementation "details" are not specified. For instance, in Equation (6), the BLEU function is defined at the sentence level while in the actual BLEU metric is defined at the corpus level. The authors should have specified which approximation of BLEU that have used. The connection between structure prediction (especially in NLP) and reinforcement learning has been made since 2005 (see for instance [1], [2] or, more recently, [3]). It is, today, at the heart of most dependency parsers (see, among others, the work of Goldberg) that have deal with similar issues for a long time. The related work section should not (and most not) be only about neural network works. The MT system used in the experiments is far behind the performance achieved by the best systems of the WMT campaign: the best system (a "simple" phrase-base system as I remember) achieves a BLEU score of ~20 for En -> De and 35 for En -> Fr while the scores reported by the authors are respectively ~15 and ~30. I wonder if the authors would have observed similar gains had they considered a better baseline system. More generally, most of the reported experimental results are quite disturbing. Figure 3.a shows that increasing the beam actually degrades translation quality. This counter-intuitive results should have been analyzed. Figure 3.b indicates that taking into account the predicted value of the expected BLEU actually hurts translation performance: the less weight the predicted score gets in the linear combination, the better the translation. [1] Learning as Search Optimization: Approximate Large Margin Methods for Structured Prediction, Hal Daumé III and Daniel Marcu, ICML'05 [2] Searched-based structured prediction, Hal Daumé III, John Langford and Daniel Marcu, ML'09 [3] Learning to Search Better than Your Teacher, Kai-Wei Chang, Akshay Krishnamurthy, Alekh Agarwal, Hal Daumé III and John Langford

Reviewer 3



Decoding with Value Networks for Neural Machine Translation At various places, you mention the similarity between your approach and AlphaGo, which also scores actions using a linear combination of a scores from conditional probability model and scores that estimate the future reward that an action will lead to. However, I wish this discussion was a bit more precise. Namely, you should mention Monte Carlo Tree Search and how it is different than your approach. MCTS has various methods for balancing exploration vs. exploitation that you ignore when you sample your trajectories from your policy. These may have been useful. Your discussion of related work should include the broader selection of RL methods that have been used in NMT over the past couple of years. For example, it would be helpful for readers if you mentioned actor-critic approaches and described how your work is different. You definitely need to improve the exposition regarding the semantics for the policy that the value network is computed with respect to. You say that it estimates the expected reward under a policy. However, you never state formally what this distribution is. Since you use the avg bleu score, it seems like you’re defining it as the distribution defined by running beam search to get a set of candidate predictions and then uniformly sampling from these. Why is this the right distribution to be using? You could, for example, just sample from the conditional distribution of y_t given the y's before it. I was very confused why you chose avg bleu and not max bleu in line 165. Shouldn’t the score of a word be the maximum possible reward achievable if we choose that word? There is a nice line of related work in the parsing literature on defining what these look-ahead oracles should be. See Goldberg and Nivre [1] and follow-on work. [1] Goldberg and Nivre. A Dynamic Oracle for Arc-Eager Dependency Parsing. 2012. Minor comments line 156: "Temple Difference" line 149: it would be helpful to remind the reader what h is (same for c in line 142) line 152: by softmax do you mean sigmoid? Also, why is this a good idea? line 163: it’s not really an expectation: it’s the deterministic set output by beam search citations: you should use citet in the natbib package to reference papers as elements in the sentence. For example, rather than saying ‘[1] do xyz’ you should say ‘author1 et al. [1] do xyz' references section: often you cite the arxiv version of a paper that has been accepted to a conference